# The Use of Scoring Hip Osteoarthritis with MRI as an Assessment Tool for Physiotherapeutic Treatment in Patients with Osteoarthritis of the Hip

**DOI:** 10.3390/jcm11010017

**Published:** 2021-12-21

**Authors:** Agnieszka Lewińska, Piotr Palczewski, Krzysztof Piłat, Andrzej Cieszanowski, Witold Rongies

**Affiliations:** 1Chair of Clinical Physiotherapy, Faculty of Rehabilitation, Jozef Pilsudski University of Physical Education in Warsaw, 00968 Warsaw, Poland; 21st Department of Clinical Radiology, Faculty of Medicine, Medical University of Warsaw, 02004 Warsaw, Poland; sounded@wp.pl (P.P.); pilat.krzysztof@gmail.com (K.P.); 32nd Department of Clinical Radiology, Faculty of Medicine, Medical University of Warsaw, 02097 Warsaw, Poland; andrzej.cieszanowski@wum.edu.pl; 4Department of Rehabilitation, Faculty of Medical Sciences, Medical University of Warsaw, 02109 Warsaw, Poland; rongies@interia.pl

**Keywords:** SHOMRI, MRI, hip, osteoarthritis, physiotherapy

## Abstract

Rehabilitation programs are considered effective at reducing the impact of osteoarthritis (OA) of the hip; however, studies using reliable measures related to OA biomarkers to assess the effects of rehabilitation are lacking. The objective of this study was to investigate whether an MRI-based (Magnetic Resonance Imaging-based), semi-quantitative system for an OA severity assessment is feasible for the evaluation of the structural changes in the joint observed during a long-term physiotherapy program in patients with hip OA. The study group consisted of 37 adult OA patients who participated in a 12-month physiotherapy program. The Scoring hip osteoarthritis with MRI (SHOMRI) system was used to evaluate the severity of structural changes related to hip OA. Hip disability and the osteoarthritis outcome score (HOOS) and the core set of performance-based tests recommended by Osteoarthritis Research Society International were used for functional assessment. SHOMRI showed excellent inter- and intra-rater agreement, proving to be a reliable method for the evaluation of hip abnormalities. At the 12-month follow-up no statistically significant changes were observed within the hip joint; however, a trend of structural progression was detected. There was a negative correlation between most of the SHOMRI and HOOS subscales at baseline and the 12-month follow-up. Although SHOMRI provides a reliable assessment of the hip joint in patients with OA it showed a limited value in detecting significant changes over time in the patients receiving physiotherapy over a 12-month period.

## 1. Introduction

Osteoarthritis (OA) affects more than 303 million people worldwide, and by 2032 the proportion of the population aged 45 years and older with doctor-diagnosed OA at any location is estimated to reach the level of 29.5% [1,2]. In the European population, the prevalence of radiographic OA of the hip in middle-aged women and men varies, but based on the data available it can be estimated to be between 15.9–18.6% and 14.1–27.3%, respectively [3,4]. There are differences between countries in the prevalence of OA, but its burden is undeniably and consequently increasing [5]. OA is predicted to become the greatest cause of disability globally, as its associated symptoms result in a substantial decrease in function and loss of working capacity [6]. A strong impact on individual and population health is also linked with enormous social and medical costs. The socioeconomic burden of osteoarthritis is estimated at between 1% and 2.5% of the gross domestic product in developed countries [7].

A growing understanding of OA pathogenesis results in an increasing range of pathologic processes that may be targeted to prevent disease progression. However, the disease management of hip OA still largely relies on symptomatic treatment and total hip arthroplasty for patients that continue to have persistent pain despite treatment [8,9]. In the initial stages of hip OA, the inclusion of conservative non-pharmacological strategies in treatment regimen is recommended [10]. Rehabilitation programs implemented at early stage of the disease are considered to be effective at reducing the impact of OA; they manage to lessen the pain, increase quality of life and physical activity, and decrease the risk of arthroplasty [11,12]. On the other hand, treatment guidelines recommending rehabilitation for people with symptomatic hip osteoarthritis are based on limited evidence [13]. Most previous research included outcome measures that were mainly concerned with patient-reported parameters of intensity of symptoms, quality of life, and functional status accompanied by functional tests [13,14]. Such clinical outcomes are unfortunately unable to clarify whether the intervention only alleviates symptoms or modifies the disease process at a joint tissue level. To answer this question, the effects of rehabilitation should be assessed using reliable and validated outcome measures related to OA biomarkers, similarly to other therapeutic approaches currently being tested. Only when a proper assessment methodology is constructed, can the effects of rehabilitation be reliably compared to other OA treatments.

To objectively assess the effect of an intervention on the disease process, imaging techniques are used along with clinical parameters [15]. Although magnetic resonance imaging (MRI) is not routinely employed for OA diagnosis, in recent years it has been increasingly used for research and clinical trials. The advantages of MRI, besides providing a direct visualization of articular cartilage, are related to an excellent soft tissue contrast that allows a whole-joint assessment in OA [16]. The recognition of OA as a disease that affects all structures of the joint resulted in the introduction of scoring systems for the assessment of degenerative changes across the hip joint [17,18,19]. MRI-based semi-quantitative systems of OA severity assessment are recommended for clinical trials by the Osteoarthritis Research Society International (OARSI) [20]. The scoring hip osteoarthritis with MRI (SHOMRI) is an exemplary tool with high construct validity and inter-observer agreement that evaluates articular cartilage loss, bone marrow edema pattern (BMEP), subchondral cysts, labral abnormalities, joint effusion, loose bodies, and ligamentum teres abnormalities [19,21]. Using arthroscopic correlation, SHOMRI grading of the hip proved to be a valid and precise method to assess chondrolabral abnormalities [21]. Additionally, SHOMRI has been used for the longitudinal assessment of OA progression, and a correlation between some of the assessed parameters (namely BMEP and subchondral cysts) and functional evaluation parameters has been demonstrated [22].

The purpose of this study was to evaluate whether SHOMRI is applicable as an outcome measure in physiotherapeutic intervention, in particular in terms of detecting structural changes over time and linking them to functional parameters. We hypothesized that within the course of a long-term physiotherapy program changes in some of the MRI parameters may occur and that SHOMRI may be a feasible tool for future research on the effects of physiotherapy on hip OA. To verify this hypothesis, we evaluated the reliability of SHOMRI, assessed longitudinally structural MRI parameters and functional parameters of the hip joint, and subsequently investigated the relationship between them.

## 2. Materials and Methods

### 2.1. Study Design

The trial was conducted prospectively. General plan of the work consisted of group recruitment, physiotherapy process with periodic patient evaluation, and the analysis of the parameters obtained. After patients’ enrollment the inclusion eligibility was assessed with a preliminary questionnaire. Subsequently, a physiotherapy program was implemented, and an assessment of the participants was conducted. The subjects participated in three rounds of 3-week physiotherapist-supervised treatment at the rehabilitation outpatient clinic, with two 5-month intervals of an unsupervised home-based maintenance program in between. Each round of supervised physiotherapy consisted of five therapeutic sessions per week. The supervised physiotherapy program focused on pain reduction, active range of motion improvement, and obtaining proper muscle control. Each session lasted approximately 90 min and consisted of hip joint traction procedure followed by hip suspension exercises as well as lower extremities muscle strengthening and proprioception training. Additionally, transcutaneous electrical nerve stimulation (TENS) was used (Multitronic MT-6, EiE, Otwock, Poland). The duration of the intervention period was 12 months.

Whole hip structural changes were studied as well as hip-related functional status. The functional assessment was performed four times, at the baseline and after each round of physiotherapy. It was conducted using a self-reported questionnaire and the assessor-observed performance-based tests. The hip joint structural evaluation with a semi-quantitative MRI-based scoring system was performed two times, at the baseline and after 12 months.

The study obtained the approval of the Committee on Bioethics of the Medical University of Warsaw, Poland. The trial was registered on The Australian New Zealand Clinical Trials Registry (ANZCTR) with the number ACTRN12621000489897. All patients provided a written informed consent for the participation in the study. Enrollment of patients and completion of study is demonstrated in the flow chart in Appendix A.

### 2.2. Participants

The study group consisted of 37 patients of both genders who met the inclusion criteria. The group was selected among patients of Department of Rehabilitation of Central Teaching Clinical Hospital of Medical University of Warsaw. Consecutive subjects were invited to participate in the trial. Patients had the right to withdraw from the study at any time with no need to provide a reason for withdrawal.

At the baseline descriptive information regarding the overall health status, medication use, co-morbidities, duration of hip OA symptoms, and demographic factors including age, gender, body mass index (BMI), and employment status, were obtained by questionnaire. Disease severity was assessed on hip radiographs using the Kellgren–Lawrence grading system (K-L) [23]. Overall average hip pain during the last week was assessed using a 0–10 numerical rating scale (NRS).

Eligibility criteria for participants included: over 18 years of age; hip osteoarthritis fulfilling American College of Rheumatology (ACR) classification criteria [24]; hip joints weight bearing plain radiography within 6 months; and written informed consent provided.

Exclusion criteria included: contraindications for MRI, physical therapy treatment or physical activity; systemic arthritic conditions or diseases and lesions within the musculoskeletal system (other than osteoarthritis of the hip) that could significantly affect the condition of the hip joint and the patient’s functional capabilities; prior hip surgery or lower extremity joint replacement; intra-articular corticosteroid injection or oral steroid or nonsteroidal anti-inflammatory drugs (NSAID) chronic use within six months; viscosupplementation within six months; prior cerebral vascular accident or other neurological disorders affecting sensorimotor functions; history of myocardial infarction; history of cancer; and general poor health status.

### 2.3. Structural Outcome Measures

Hip joint assessment was conducted using MRI and performed on a 1.5 T scanner (Avanto, Siemens, Erlangen, Germany using a spine coil integrated into the table and surface body coil. The MRI protocol included: T1-weighted TSE sequence, PD TSE sequence with fat saturation, PD SPACE (3D TSE) sequence, and T2-weighted double-echo 3D sequence. All images were acquired in coronal plane with a field of view covering both hips. Detailed MRI protocol is presented in Table 1.

The MRI images were reviewed independently by two radiologists with more than 5 years of experience in musculoskeletal radiology (K.P. and P.P.). The assessors were not informed of the clinical and functional information other than sex and age. Any disagreements were resolved during a final consensus reading session with both readers assessing the images together.

The severity of degenerative changes in both hips was analyzed using the SHOMRI evaluation system. Articular cartilage lesions, bone marrow edema, and subchondral cysts were scored in six femoral and four acetabular subregions, and addedsubsequently for a subscore specific to each feature. Labral abnormalities were scored in four subregions and added for a subscore. Paralabral cysts, intra-articular bodies, effusion and/or synovitis, and ligamentum teres abnormalities were individually scored. All subscores were averaged together to create a total score [19].

Time needed to score both hips was recorded with a stopwatch for each reading.

To assess the reliability of SHOMRI scoring system, the results of the baseline evaluation were used. For inter-reader analysis the images were assessed independently by two readers (P.P. and K.P.) not informed of each other’s results. For intra-reader analysis the images were assessed twice by each reader in the 6 months interval. The readers were not informed of their previous results.

In patients that underwent both baseline and follow-up MRIs, the progression of degenerative changes was assessed by comparing SHOMRI scores in each category and also by consensus reading of MRI images by both radiologists (K.P. and P.P.) to look for changes not reflected by change in SHOMRI scores. Based on this analysis the patients were divided into progressors and non-progressors groups.

### 2.4. Functional Outcome Measures

The functional assessment was conducted using a self-reported questionnaire and the assessor-observed performance-based tests. As a patient-reported outcome measure (PROM), the hip dysfunction and osteoarthritis outcome score (HOOS) was used to assess hip-related function over the previous week of activity. HOOS is composed of 5 separately scored subscales and provides an estimate of each subject’s symptoms, pain, activities of daily living limitations (ADL), sport and recreation function (SR), and quality of life assessment (QOL). A percentage score ranging from 0 to 100 was calculated for each subscale where 100 indicates no disability and 0 indicates severe disability. [25]

In addition to PROM, OARSI-recommended physical function tests were used to assess physical performance. The set of performance-based tests consisting of the 30 s chair stand test (30secCST), 40-meter fast-paced walk test (40mFPWT) and stair climb test (SCT) relates to the ability of walking, climbing, changing body position, and moving around. The tests were assessed by counting, time, and speed measure [26]. Performance-based tests were assessed by one physiotherapist with 5 years of experience.

### 2.5. Statistical Analysis

Prior to analysis, data were cross-checked for missing values and outliers. The missing items (3 values in the HOOS self-reported questionnaire in 3 different patients) were replaced with the average of the observed data for that variable in other patients according to the mean substitution approach. The Shapiro–Wilk normality test was used to verify the distribution of the data. Descriptive statistics were used to describe the baseline characteristics of the sample. Discrete variables were described as median and interquartile range (IQR), and categorical variables were described by patient counts and percentages. Since the data were not normally distributed, the Mann–Whitney U test (Z) was used to compare the differences between the groups. To examine the differences between the structural outcome in two consecutive time points, Wilcoxon signed-rank tests (Z) were used. Categorical variables were evaluated for differences with McNemar’s χ^2^ test for paired data. Krippendorff’s alpha (α) reliability coefficient was used for determining inter-rater and intra-rater reliability, as the data analyzed were collected in an ordinal and dichotomous scale. Possible Krippendorff’s α values range from 0 to 1.0, where 0.0 means no agreement, and 1.0 equates to perfect agreement. A cutoff threshold value of 0.8 is suggested as a marker of good reliability [27]. Correlations between imaging and functional parameters were assessed by using Spearman’s rank correlation coefficient (r) for ordinal variables and by point biserial correlation coefficient (r_pb_), where the data analyzed were presented in ordinal and dichotomous scale. A statistical significance level of 0.05 was regarded for all tests. The statistical analysis was conducted using Statistica PL version 13.3 (TIBCO Software Inc., Palo Alto, CA, USA) and Microsoft Excel (Microsoft Corporation, Redmond, WA, USA).

## 3. Results

### 3.1. Study Group

A total of 54 patients with hip OA were screened to determine eligibility, with 37 included, and 24 eventually completed the intervention. No differences were registered in the study group in terms of structural degeneration of the hip joint due to age, sex, occupation, or BMI. Patients baseline characteristics are summarized in Table 2.

### 3.2. Reproducibility

Krippendorff’s alpha reliability coefficients for inter- and intra-reader analysis were excellent for the SHOMRI total and all subscales assessed. The lowest values were observed for inter-reader agreement of joint effusion assessment. Table 3 presents the outcome of analysis conducted.

The average time required to score both joints for the first reading of the baseline study for P.P. and K.P. was 21 and 24 min, respectively. Detailed numbers of the average time required to score both joints are demonstrated in Appendix A.

### 3.3. Radiological Evaluation

Baseline MRI evaluation revealed statistically significant differences between symptomatic and asymptomatic joints in the total SHOMRI score. The differences were also observed in the BMEP sub-score at the baseline as well as at the 12-month follow-up (Table 4).

In the study group, no statistically significant changes in any of SHORMI subscales were observed during the 12-month follow-up. However, on an individual basis, two patients showed a progression of cartilage defects reflected by a change in the SHORMI score from 1 to 2, and one patient showed a progression of subchondral cysts similarly reflected by an increase in the SHORMI score. In three patients, enlargement of cartilage defect area was observed; however, since those were already full-thickness defects, it did not affect SHORMI score (Figure 1); in one of those patients the area of bone marrow edema increased as well. In one patient, the paralabral cyst increased substantially. All of those patients (*n* = 7) were labeled “progressors”, in contrast to the remaining patients (*n* = 17) that showed no change in MRI (“non-progressors”) (Table 5). This division was subsequently used in further analysis.

### 3.4. Correlation

There was a negative correlation between the total SHOMRI score and HOOS demonstrated both at the baseline and at the 12-month follow-up in the study group. Moreover, SHOMRI ordinal subscales showed a relationship between low and moderate negative correlation with most of the HOOS domains. The greatest magnitude of significant correlation was shown for symptoms and subchondral cysts, while the lowest significant association was shown for pain and subchondral cysts. No correlation has been observed for SHOMRI- and performance-based tests (Table 6).

There was no correlation between SHOMRI and HOOS in the group of progressors, while the outcomes of non-progressors suggested a relationship between SHOMRI cartilage and several of the HOOS features (Table 7).

## 4. Discussion

The purpose of this study was to evaluate whether SHOMRI is applicable as an outcome measure in long-term physiotherapy intervention, in particular in terms of detecting structural changes over time and linking them to functional parameters.

In the literature there is relatively little research on osteoarthritis that concerns the studies of the hip joint, and the predominant focus on the knee reduces the possibilities of a broad discussion. There are limited trials that objectively investigate structural changes in those completing physiotherapy programs. In this field, there has not been significant research conducted in recent years regarding hip joint imaging [28,29,30] or the long-term effects of physiotherapeutic treatment [31,32,33].

Excellent α values for the SHOMRI total and all assessed subscales indicate its usefulness as a measurement tool. The results obtained in this study are comparable to previously reported modest to excellent reproducibility parameters [19,21,22,34]. Interpretation of the results is consistent for most features, and slight variations in the values of the coefficients may arise from different statistical methods used. For example, Lee et al., hypothesized that modest values obtained in their study may have been partly related to the low frequency of abnormalities as Kappa values may, in such circumstances, underestimate the agreement, leading to low kappa despite high proportional agreement [19,35]. As the SHOMRI outcomes are presented with the use of an ordinal and dichotomous scale, assessing its reproducibility requires a targeted approach that prevents considerable underestimation of the measurements’ true reliability. Krippendorff’s alpha coefficient was used in this study for determining reliability of measurements due to its high flexibility regarding the measurement scale. Even though Krippendorff’s alpha was not originally described as a method for intra-rater reliability assessment, such analysis was conducted according to the suggestion of Zapf et al. [36], as in the present study there were similarly no systematic differences in the way the parameters were assessed.

The mean times of the assessment recorded in the present study were similar to those previously reported in the literature. Lee et al. reported that scoring a single hip required 9 min 06 s ± 4 min 28 s, while times required to score both hips in our study ranged from 15 min 12 s ± 3 min 33 s to 24 min 10 s ± 8 min 22 s) [19]. Even taking into account a learning curve visible, as shorter times are required to score both hips at the subsequent readings, this approach may be considered time-consuming, which may be an obstacle in everyday clinical practice and suggests that its application may be rather more beneficial in the area of research.

The fact that no statistically significant changes were found regarding the progression of hip abnormalities assessed by SHOMRI does not support its use to assess the effectiveness of physiotherapeutic intervention in terms of whole-joint structural changes. The sensitivity of the tool may not be sufficient enough to detect the development of OA within the joint if it is not sufficiently pronounced. On the other hand, it needs to be stated that the sensitivity of SHOMRI in terms of detecting progression depends on the cut-off points adopted. Lee et al. admitted that the number of the point-scale increments and regions may affect the systems’ sensitivity to interval change [19]. With the use of a high-resolution, three-dimensional sequence in the present study it was possible to observe subtle structural changes that were not sufficiently marked to be acknowledged by SHOMRI; thus, these could have remained potentially unnoticed in previous reports. Considering that in the study conducted by Schwaiger et al. [22] and Gallo et al. [37], only minimal differences were perceived over 1.5 years observation, it may be hypothesized that a longer follow-up period might be necessary to pick up the changes.

Interestingly, the trend of structural progression was observed exclusively among patients declaring occupations of a physical nature. This appears to be consistent with the findings from an umbrella review of systematic reviews by Schram et al., who found that occupational physical tasks related to forces exerted on the hip were associated with an increased risk of hip OA [38]. It is worth noticing that the level of occupational activity was the only feature distinguishing both groups of progressors and non-progressors. There were also no differences observed in other parameters assessed among office and physical workers at the baseline.

Although there were no differences in K-L grade between symptomatic and asymptomatic joints in patients accepted in the study, the general perception of symptoms of the hip joint reported by patients upon enrollment was found to be significantly associated with BMEP. When specifying the hip joint with predominant ailments, the one with a higher SHOMRI BMEP sub-score was most often pointed out, which can be considered consistent with findings by Taljanovic et al., showing that the amount of BMEP correlates significantly with hip pain [39].

The occurrence of marked correlation between the features of structural and functional assessment suggests that SHOMRI has potential use in studies that focus on the clinical manifestation of the hip joint disease. Such a relationship has been confirmed before in several studies where HOOS was used as a functional outcome measure [19,22,40]. In the presence of evidence from our study it could be safe to assume that SHOMRI in general shows a correlation of moderate magnitude and HOOS. However, the strength of the correlation in particular domains and even its presence may vary greatly, depending on the studied population or the time point, when assessed in the course of physiotherapeutic intervention. In our study, the manifestation of correlation as well as its magnitude varied when assessed in groups and was demonstrated only among the group of non-progressors. There was also discordance recognized in the correlation between structural and functional parameters when assessed at the baseline and 12-month follow-up. Although no straightforward conclusion can be drawn from these results, they may, however, suggest that simple dependency between structural and functional features cannot be fully relied upon and needs further investigation.

It is also worth mentioning that the link documented between SHOMRI and PROM in all of the studied population was not confirmed in the assessor-observed performance-based tests. Although in theory they are meant to reflect the actual functional status of the patients, Tolk et al., suggested that OARSI-recommended performance-based measures may not target the exact same domain of physical function as PROM [41]. Thus, hypothetically structural changes assessed with SHOMRI may not affect the quantitative result of performance-based tests. However, dependencies between the methods of functional assessment are beyond the scope of this paper, and extended assessment is needed to further elucidate this lack of relationship.

Several limitations of this study should be highlighted. The main factor that could have influenced the results was the small sample size of 37 subjects. It is possible that a larger study group could expose changes not being detected. Another limitation is a lack of a control group, but for ethical reasons, it was not possible to create a symptomatic control group that would not receive treatment for the period of one year. It is also worth considering that the follow-up period may have been too short to allow for the evolution of hip joint abnormalities.

In conclusion, it should be emphasized that SHOMRI has been characterized by excellent reliability and thus may be used to quantify the structural changes of the hip joint and lead to a better understand of their contributions to hip function. It seems to be an acceptable tool to draw some connection between the structural and functional parameters of symptomatic hips in the general population of OA patients; however, its usefulness in the assessment of different sub populations remains unclear. Finally, its performance in terms of detecting significant changes over time appears to be insufficient. Further studies are needed to assess the relationships between MRI features and the clinical symptoms of hip OA, especially over time, as well as to evaluate the relevance of MRI features as predictors of the progression and response to treatment.

## Figures and Tables

**Figure 1 jcm-11-00017-f001:**
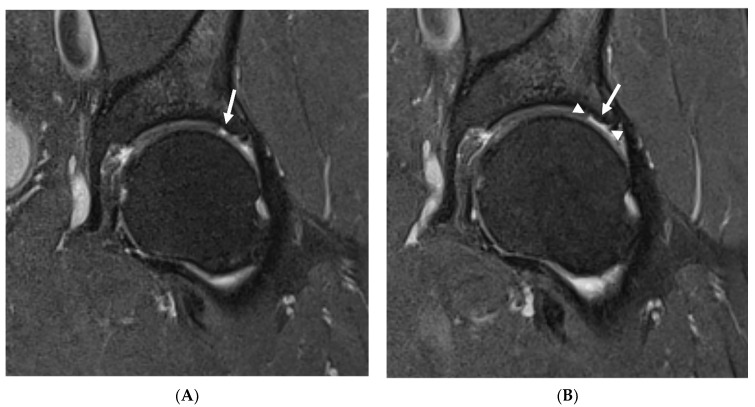
Progression of chondral defects missed by SHOMRI. Baseline MRI (**A**) showing a full thickness cartilage defect (arrow) corresponding to SHOMRI grade 2 in a superolateral acetabular region in a 60-year-old patient. In a follow-up study (**B**), the defect covers a larger area (arrowheads show the margins of the defect), which still corresponds to SHOMRI grade 2.

**Table 1 jcm-11-00017-t001:** MRI protocol.

Sequence	TR [ms]	TE [ms]	Flip Angle (°)	FOV [mm]	Slice Thickness/Gap [mm]	Matrix	Number of Averages	EchoTrain Lenght	Number of Slices	Acquisition Time [min]
PD SPACE cor iso	1200	42	160	262 × 400	0.8/0.0	448 × 294	2	153	96	7:30
T2 de3D cor iso	18	6.5	25	262 × 400	0.8/0.0	512 × 316	1	1	96	7:30
PD TSE FS cor	3490	44	150	210 × 330	3.0/0.3	512 × 316	2	10	25	6:20
T1 TSE cor	744	22	180	210 × 320	3.0/0.3	512 × 336	2	2	25	2:15

Abbreviations: TR: repetition time; TE: echo time; FOV: field of view.

**Table 2 jcm-11-00017-t002:** Baseline demographics.

Included (*n* = 37)	
Age (years; median (IQR))	58.00 (12.00)
BMI (median; IQR)	25.48 (4.52)
Sex:	
(female; *n* (%))	21 (56.8%)
(male; *n* (%))	16 (43.2%)
Occupational activity:	
(sedentary; *n* (%))	23 (62.2%)
(active; *n* (%))	14 (37.8%)
Symptomatic joint:	
(left; *n* (%))	18 (48.65%)
(right; *n* (%))	19 (51.35%)
NRS:	
(symptomatic joint; median (IQR))	4.00 (4.00)
(asymptomatic joint; median (IQR))	1.00 (3.00)
KL grade:	
(symptomatic joint; *n* (%))	1 = 5 (13.51%)
	2 = 15 (40.54%)
	3 = 14 (37.84%)
	4 = 3 (8.11%)
(asymptomatic joint; *n* (%))	1 = 7 (18.92%)
	2 = 26 (70.27%)
	3 = 4 (10.81%)
	4 = 0 (0.00%)

Abbreviations: BMI: body mass index; NRS: numerical rating scale; KL: Kellgren–Lawrence grading system; variables are expressed as median, inter-quartile range (IQR), patient counts, and percentages.

**Table 3 jcm-11-00017-t003:** Inter-reader and intra-reader agreement.

MRI Parameter	α	α_PP_	α_KP_
SHOMRI total	0.95	0.96	0.99
SHOMRI Cartilage	0.90	0.92	0,97
SHOMRI BMEP	0.97	1.00	1.00
SHOMRI Subchondral cysts	0.95	0.95	0.97
SHOMRI Labrum	0.96	0.92	0.97
SHOMRI Paralabral cysts	1.00	1.00	1.00
SHOMRI Effusion/synovitis	0.86	0.93	0.93
SHOMRI Intraarticular bodies	1.00	1.00	1.00
SHOMRI Ligamentum teres	0.97	0.97	1.00

Abbreviations: SHOMRI: scoring hip osteoarthritis with MRI; BMEP: bone marrow edema pattern; α: Krippendorff’s alpha reliability coefficient; α_PP_: Krippendorff’s alpha reliability coefficient for 1st reader (P.P.); α_KP_: Krippendorff’s alpha reliability coefficient for 2nd reader (K.P.).

**Table 4 jcm-11-00017-t004:** Differences between symptomatic and asymptomatic joints in SHOMRI at baseline and 12-month follow-up.

SHOMRI	Baseline *n* = 37				12 Months *n* = 24			
	S	A	Z	*p*	S	A	Z	*p*
SHOMRI total	10.00 (16.00)	6.00 (6.00)	**−2.03**	**0.043**	6.00 (3.00)	5.00 (3.00)	−1.89	0.058
Cartilage	5.00 (4.00)	5.00 (3.00)	−1.71	0.087	0.00 (2.50)	0.00 (0.00)	−1.66	0.096
BMEP	0.00 (3.00)	0.00 (0.00)	**−2.92**	**0.003**	0.00 (2.50)	0.00 (0.00)	**−2.65**	**0.008**
Subchondral cysts	0.00 (3.00)	0.00 (1.00)	−1.46	0.143	3.50 (5.50)	1.50 (2.00)	−1.41	0.158
Labrum	3.00 (6.00)	2.00 (2.00)	−1.80	0.072	10.50 (16.50)	6.50 (5.00)	−1.83	0.067

Abbreviations: S: symptomatic; A: asymptomatic; Z: Mann–Whitney U test; values are expressed as median (inter-quartile range). Bold font highlights significant results.

**Table 5 jcm-11-00017-t005:** MRI outcomes of symptomatic hip at baseline and 12-month follow-up in a group of SHOMRI progressors and non-progressors.

	Progressors		Non-Progressors	
Parameter	Baseline	12 Months	Baseline	12 Months
	*n* = 7	*n* = 7	*n* = 17	*n* = 17
SHOMRI total	14.00 (28.00)	16.00 (28.00)	8.00 (8.00)	8.00 (10.00)
SHOMRI Cartilage	6.00 (8.00)	8.00 (9.00)	5.00 (3.00)	5.00 (3.00)
SHOMRI BMEP	0.00 (7.00)	0.00 (7.00)	0.00 (1.00)	0.00 (2.00)
SHOMRI Subchondral cysts	0.00 (3.00)	2.00 (5.00)	0.00 (0.00)	0.00 (0.00)
SHOMRI Labrum	6.00 (6.00)	6.00 (6.00)	2.00 (4.00)	2.00 (4.00)
SHOMRI Paralabral cysts	1.00 (1.00)	1.00 (1.00)	0.00 (1.00)	0.00 (1.00)
SHOMRI Effusion/synovitis	0.00 (0.00)	0.00 (1.00)	0.00 (0.00)	0.00 (0.00)
SHOMRI Intraarticular bodies	0.00 (0.00)	0.00 (0.00)	0.00 (0.00)	0.00 (0.00)
SHOMRI Ligamentum teres	0.00 (1.00)	0.00 (1.00)	0.00 (1.00)	0.00 (1.00)

Values are expressed as median (inter-quartile range).

**Table 6 jcm-11-00017-t006:** Correlation analysis among symptomatic hip MRI and functional parameters at baseline and 12-month follow-up.

Parameter	Time Point	*n*	HOOS Total	Pain	Symptoms	ADL	SR	QOL	40-mFPWT	30 s CST	SCT
SHOMRI total	I	37	**−0.41 (0.011)**	−0.31 (0.057)	**−0.41 (0.012)**	**−0.39 (0.016)**	**−0.39 (0.015)**	**−0.38 (0.021)**	0.18 (0.276)	−0.06 (0.710)	−0.10 (0.570)
	IV	24	**−0.50 (0.012)**	**−0.49 (0.015)**	**−0.62 (0.001)**	**−0.52 (0.009)**	**−0.48 (0.018)**	**−0.41 (0.048)**	−0.15 (0.474)	0.22 (0.292)	−0.06 (0.791)
Cartilage	I	37	**−0.49 (0.002)**	**−0.39 (0.018)**	**−0.48 (0.002)**	**−0.47 (0.003)**	**−0.46 (0.004)**	**−0.42 (0.009)**	−0.25 (0.130)	−0.09 (0.578)	0.16 (0.346)
	IV	24	**−0.44 (0.030)**	**−0.44 (0.031)**	**−0.51 (0.010)**	**−0.50 (0.013)**	**−0.45 (0.029)**	−0.36 (0.083)	−0.19 (0.375)	0.29 (0.161)	0.02 (0.937)
BMEP	I	37	**−0.38 (0.021)**	−0.29 (0.076)	**−0.37 (0.023)**	−0.36 (0.029)	**−0.40 (0.015)**	−0.31 (0.064)	−0.18 (0.293)	0.23 (0.892)	0.08 (0.632)
	IV	24	**−0.54 (0.006)**	**−0.53 (0.007)**	**−0.59 (0.002)**	**−0.53 (0.007)**	**−0.49 (0.016)**	**−0.49 (0.015)**	−0.34 (0.100)	0.24 (0.257)	0.03 (0.897)
Subchondral cysts	I	37	**−0.42 (0.010)**	**−0.35 (0.033)**	**−0.49 (0.002)**	−0.32 (0.052)	**−0.35 (0.034)**	**−0.41 (0.013)**	−0.03 (0.854)	0.05 (0.786)	0.08. (0.621)
	IV	24	−0.28 (0.190)	−0.24 (0.260)	−0.39 (0.060)	−0.24 (0.250)	−0.21 (0.334)	−0.29 (0.165)	−0.03 (0.893)	0.34 (0.098)	−0.09 (0.689)
Labrum	I	37	−0.11 (0.505)	−0.06 (0.702)	−0.06 (0.712)	−0.12 (0.468)	−0.11 (0.506)	−0.15 (0.369)	−0.37 (0.826)	−0.57 (0.657)	−0.05 (0.779)
	IV	24	−0.32 (0.124)	−0.34 (0.107)	**−0.48 (0.016)**	−0.37 (0.078)	−0.32 (0.132)	−0.20 (0.345)	−0.01 (0.967)	0.12 (0.559)	−0.22 (0.304)
Paralabral cysts *	I	37	−0.01 (0.928)	0.07 (0.665)	−0.01 (0.933)	−0.05 (0.786)	−0.06 (0.729)	−0.02 (0.925)	−0.01 (0.936)	−0.00 (0.970)	0.03 (0.871)
	IV	24	**−0.40 (0.048)**	−0.35 (0.081)	**−0.44 (0.028)**	**−0.41 (0.040)**	−0.37 (0.069)	−0.31 (0.133)	−0.06 (0.760)	−0.04 (0.845)	−0.07 (0.746)
Effusion/synovitis *	I	37	−0.30 (0.070)	−0.24 (0.156)	−0.30 (0.070)	−0.23 (0.177)	**−0.33 (0.045)**	−0.31 (0.064)	−0.03 (0.843)	−0.00 (0.997)	−0.05 (0.826)
	IV	24	−0.37 (0.069)	−0.32 (0.114)	−0.34 (0.091)	−0.28 (0.179)	**−0.40 (0.047)**	**−0.41 (0.017)**	−0.02 (0.924)	−0.−4 (0.848)	−0.10 (0.622)
Intraarticular bodies *	I	37	N/A	N/A	N/A	N/A	N/A	N/A	N/A	N/A	N/A
	IV	24	N/A	N/A	N/A	N/A	N/A	N/A	N/A	N/A	N/A
Ligamentum teres *	I	37	−0.32 (0.051)	−0.29 (0.082)	**−0.35 (0.031)**	−0.28 (0.093)	−0.27 (0.107)	−0.32 (0.056)	−0.28 (0.098)	−0.11 (0.522)	0.17 (0.314)
	IV	24	**−0.48 (0.015)**	**−0.46 (0.020)**	**−0.47 (0.018)**	**−0.43 (0.030)**	**−0.44 (0.027)**	**−0.47 (0.017)**	**−0.45 (0.022)**	−0.28 (0.169)	**−0.48 (0.014)**

Abbreviations: HOOS: hip dysfunction and osteoarthritis outcome score; ADL: activities of daily living; SR: sport and recreation; QOL: quality of life; 30-second CST: 30-second chair stand test; 40mFPWT: 40 m fast-paced walk test; SCT: stair climb test; I: baseline; IV: 12 months; Values are expressed as Spearman’s correlation rank coefficient r (p) or point biserial correlation coefficient * r_pb_ (p). Bold font highlights significant results.

**Table 7 jcm-11-00017-t007:** Correlation analysis among symptomatic hip SHOMRI and HOOS at baseline.

Parameter	*n*	HOOS	Pain	Symptoms	ADL	SR	QOL
Progressors							
SHOMRI total	7	−0.41 (0.355)	−0.32 (0.478)	−0.41 (0.355)	−0.54 (0.210)	−0.62 (0.140)	−0.35 (0.435)
Cartilage	7	−0.23 (0.613)	−0.16 (0.728)	−0.34 (0.452)	−0.40 (0.379)	−0.54 (0.208)	−0.42 (0.350)
BMEP	7	−0.30 (0.515)	−0.14 (0.765)	−0.42 (0.350)	−0.30 (0.515)	−0.37 (0.411)	−0.46 (0.296)
Subchondral cysts	7	−0.08 (0.867)	−0.40 (0.933)	−0.20 (0.672)	−0.18 (0.704)	−0.37 (0.417)	−0.30 (0.949)
Labrum	7	−0.16 (0.726)	−0.05 (0.907)	−0.38 (0.398)	−0.09 (0.846)	−0.09 (0.840)	−0.12 (0.799)
Paralabral cysts *	7	0.51 (0.239)	0.53 (0.220)	0.52 (0.226)	0.42 (0.345)	0.40 (0.376)	0.43 (0.333)
Effusion/synovitis *	7	0.53 (0.216)	0.35 (0.444)	0.59 (0.163)	0.43 (0.330)	0.70 (0.077)	0.37 (0.411)
Intraarticular bodies *	7	−0.46 (0.296)	−0.59 (0.166)	−0.24 (0.610)	−0.53 (0.219)	−0.38 (0.404)	−0.28 (0.545)
Ligamentum teres *	7	−0.36 (0.423)	−0.28 (0.545)	−0.44 (0.316)	−0.22 (0.633)	−0.21 (0.647)	−0.53 (0.218)
Non-progressors							
SHOMRI total	17	−0.48 (0.052)	**−0.49 (0.043)**	−0.45 (0.072)	**−0.51 (0.037)**	−0.35 (0.163)	−0.46 (0.065)
Cartilage	17	**−0.55 (0.023)**	**−0.56 (0.019)**	−0.47 (0.057)	**−0.56 (0.020)**	−0.46 (0.064)	**−0.52 (0.032)**
BMEP	17	−0.48 (0.052)	−0.44 (0.079)	−0.46 (0.064)	−0.45 (0.069)	−0.45 (0.070)	−0.41 (0.097)
Subchondral cysts	17	−0.35 (0.173)	−0.29 (0.249)	−0.48 (0.053)	−0.15 (0.558)	−0.31 (0.232)	−0.41 (0.100)
Labrum	17	−0.02 (0.932)	−0.07 (0.790)	−0.02 (0.943)	−0.13 (0.618)	−0.08 (0.758)	−0.02 (0.925)
Paralabral cysts *	17	−0.10 (0.710)	−0.08 (0.770)	−0.07 (0.790)	−0.13 (0.621)	−0.05 (0.846)	−0.13 (0.604)
Effusion/synovitis *	17	**−0.50 (0.043)**	−0.44 (0.076)	−0.37 (0.144)	−0.44 (0.073)	**−0.56 (0.018)**	−0.46 (0.065)
Intraarticular bodies *	17	N/A	N/A	N/A	N/A	N/A	N/A
Ligamentum teres *	17	−0.32 (0.213)	−0.30 (0.260)	−0.26 (0.304)	−0.36 (0.150)	−0.24 (0.360)	−0.34 (0.185)

Values are expressed as Spearman’s correlation rank coefficient r (p) or point biserial correlation coefficient (r_pb_) *. Bold font highlights significant results.

## Data Availability

The data supporting the results of this study may be available upon request.

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
