# Peer review of "The Use of Scoring Hip Osteoarthritis with MRI as an Assessment Tool for Physiotherapeutic Treatment in Patients with Osteoarthritis of the Hip"

_jcm, 2021, doi:10.3390/jcm11010017_

Round 1
Reviewer 1 Report
In general it is a well written and easy to read article and it reports on an interesting and important question. However, I think there are major limitations, which are correctly mentioned by the authors in the discussion, although somewhat understated. Crucially, evaluating a specific scoring method if structural change over a rather short period of time without control group or control method makes drawing conclusions on SHORMI in this context difficult. More specific comments below.
Abstract
- “Although SHOMRI provide reliable assessment of the hip joint in patients with OA it appears to have limited value in detecting significant changes over time”. This is of course true and an important conclusion of your work, but it should be added here that this was in the context of one-year changes in patients receiving physiotherapy (over a longer time period, or after surgical intervention, SHOMRI might show significant changes).
Introduction
- “In Europe, the prevalence of radiographic OA of the hip in middle-aged women and men is estimated at 19% and 27% respectively [3]”. This number is not based on the review that you refer to, but on one specific study from this review in Croatia, correct? It seems you picked the study with highest numbers and used that as a generalized number for Europe, which I do not think is fair (if I am wrong and these numbers are mentioned for Europe, feel free to tell me so).
- The introduction is written well and easy to read. I would recommend to shorten it somewhat. For example, the sentence “Several biomarkers related to OA symptoms and progression have been identified with MRI, including: cartilage volume, cartilage T2 relaxation times, bone marrow edema, and synovitis [15]” is true, but since you do not do anything with these biomarkers and later already name the characteristics you look at with SHOMRI, it makes the introduction longer than necessary.
- I am not very familiar with physiotherapy treatments, but in progression, imaging, and follow-up of treatments, one year (or even two years) is not considered long term.
- “We hypothesized that within the course of a long term physiotherapy program a changes in some of the MRI parameters may occur and that SHOMRI” remove “a” before “changes”.
Methods
- “The subjects participated in three 3-week physiotherapist-supervised treatment sessions at the rehabilitation outpatient clinic” what does this mean, three 3-week sessions? Three times per week for three weeks? Or nine weeks of sessions (but how many times per week then)? From the next part I gather it’s 3 times 3 weeks, but that would suggest something changed between sessions (otherwise why is it not one 9-week session), was this the case?
- “The duration of the intervention period was 12 months” I do not understand how you arrive at 12 months from the three 3-week sessions and the 5 months of home-based program.
- “The functional assessment was performed four times, at the baseline and after each round of physiotherapy” similar to the previous comments, from the text I do not understand how the schedule worked. I would suggest adding a flowchart showing when patients did what and when which evaluations were performed.
- “Based on the deterioration of structural changes observed over 12-month period, the sub-groups of progressors and non-progressors were distinguished for the purpose of statistical analysis” I would not put this here, since here the reader just wonders what the criteria were, and you repeat this later in the methods.
- You do not use educational status in your study and do not report it as part of the baseline variables, right? In that case, do not mention it here either.
- “basic blood test within 6 months” why? If you do not use it in this study, same as previous comment, do not report it in the methods either.
- What did you do when scores of the two readers did not agree?
- “The tests are assessed by counting and distance measure” what do you mean?
- “For missing items the average score was calculated and substituted for the mean overall score” what do you mean? When a patient was missing a score at one time point, it was replaced by the mean of all patients’ scores at that time point that were not missing? Or replaced by the mean of the other time points of that patient? Either way, this seems a weird way to treat missing data, why not use imputation or just leave them as missing values? How many items were missing anyway?
- “The variables were reported due to the level of measurement for the outcome variable” what do you mean? That discrete and categorical variables were reported differently? You already say this in the next sentence.
Results
- Table 2: KL grade is categorical, why not report this as n (%) for each grade? That agrees with your methods and gives a better overview of your patient group.
- Since all reproducibility coefficients are excellent, it is not necessary which parameters showed the highest and lowest values. The only exception could be the inter-observer effusion score since it is clearly lower than the rest.
- Why do you report how long scoring took, especially in such detail? From the discussion I gather that you used to evaluate SHOMRI for use in clinical practice and that makes sense, but this part is difficult to read with all the numbers. I would suggest giving average numbers very shortly in the text and putting these specific numbers in a supplementary table.
- Table 4: please also add p-values for the comparison of baseline and follow-up. How it is possible that there are no significant differences between SHOMRI scores baseline and follow-up if these numbers are correct? Are you sure the values at 12 months are correct? They do not match Table 5.
- So the patients who patients that showed enlargement of the cartilage defect area that was not reflected in a change in SHOMRI score were also classified as progressors? I do not think that is right, since this observation of enlargement is not an official SHOMRI scoring method and the whole point of your study is evaluating SHOMRI.
- Table 5: Non-progressors do show an increase in IQR for BMEP, this means there was a change (increase) in some score here, right? Why was this not a progressor then?
- Table 7: For which time point are these correlations?
- Please also evaluate changes in the other parameters, especially HOOS, over time and show results for that.
Discussion
- “The fact that no statistically significant changes were found regarding the progression of hip abnormalities assessed by SHOMRI do not support its use to assess effectiveness of physiotherapeutic intervention in terms of whole joint structural changes.” Is this really a limitation of SHOMRI, or are there simply no structural changes to pick up? If anything, you would expect deterioration as a result of OA and potentially no or less deterioration if the physical therapy is also structurally effective, or would you really expect structural improvement (regardless of scoring method)? Maybe the fact that only so few patients showed progression is a good sign. This is impossible to conclude without control group of similar hip OA patients who did not do physiotherapy though.
- For this same reason, why did you not for example measure cartilage thickness as well and see if this showed changes over time, as it is a continuous measure that is likely more sensitive to change? If we do not know IF there is any structural change it is difficult to evaluate whether SHOMRI is a good method to measure this potential change.
- “general perception of symptoms of the hip joint reported by patients upon enrollment was found to be significantly associated with BMEP. When specifying..” did you report this anywhere? In general in the discussion you mention some results that were not in the results section, which is confusing.
- Mentioning different results in correlations of HOOS with structural parameters from other studies makes this part of the discussion difficult to read. I suggest removing all these mentions (from Lee et al at line 361 until [40] at line 370) since the conclusion from all is the same and is already captured in by line 360-361 “Such a relationship has been confirmed before, in several studies where HOOS was used as a functional outcome measure”; just adding the references after this sentence is enough.
- “In our study, the manifestation of correlation as well as its magnitude varied when assessed in groups and was demonstrated only among the group of non-progressors” do you not think this is mostly because of the different group sizes?
- I would be most interested in the changes in HOOS and performance-based tests in the progressors, because in the end your goal is to evaluate the usefulness of SHOMRI in changes as a result of physiotherapy treatment, correct?
Reviewer 2 Report
The paper is interesting and generally properly prepared. It needs minor revisions regarding the style of the references list, as in actual version of the manuscript in many references some bibliographic data are lacking, the titles of the Journals in some references are presented in full version and in others as abbreviations, in some references the all words in Journal titles are written using capital letters at the beginning and in others only in first word of the title capital letter is used.
Author Response
Thank you very much for your kind review.
The references were prepared with a bibliography software package (EndNote software) as recommends by MDPI. All files for external citation management software were downloaded from PubMed.gov. The reference list was converted for BMC Medicine style which uses abbreviations for journal titles. The reason why some of the references were presented in full version is that not every journal title has an abbreviation. If the title consists of one word, or short words, then the abbreviation is usually the same as the full journal title. I changed the style for MDPI style and I believe that the citation list looks better and more uniform this way, so thank you for pointing the inconsistency.
Round 2
Reviewer 1 Report
I thank the authors for their response to prior critique. I have no further concerns.